# Hepatitis C (HCV) among Black and Latino sexual minority men (SMM) in the Southern United States: Protocol of a prospective cohort epidemiological study

Adedotun Ogunbajo[1]*, Mitchell Brooks[1], Temitope Oke[1], Omar Martinez[2], Carl Latkin[3], Kirk Myers[4], DeMarc A. Hickson[1]

1 Us Helping Us, People Into Living Inc., Washington, DC, United States of America, 2 School of Social Work, College of Public Health, Temple University, Philadelphia, PA, United States of America, 3 Department of Health, Behavior and Society, Johns Hopkins Bloomberg School of Public Health, Baltimore, MD, United States of America, 4 Abounding Prosperity Inc., Dallas, TX, United States of America

* dotunogunbajo@gmail.com

**Data Availability Statement:** There is no data to report, currently, as the data collection phase is

## Abstract

### Background

Sexual minority men (SMM) who engage in condomless anal sex and injection drug use are at increased risk for viral Hepatitis C (HCV) infection. Additionally, studies have found racial disparities in HCV cases across the United States. However, very few epidemiological studies have examined factors associated with HCV infection in HIV-negative Black and Latino SMM. This paper describes the rationale, design, and methodology of a prospective epidemiological study to quantify the HCV prevalence and incidence and investigate the individual and environmental-level predictors of HCV infection among HIV-negative, Black and Latino SMM in the Southern U.S.

### Methods

Beginning in September 2021, 400 Black and Latino SMM, aged 18 years and above, will be identified, recruited and retained over 12-months of follow-up from two study sites: greater Washington, DC and Dallas, TX areas. After written informed consent, participants will undergo integrated HIV/STI testing, including HCV, HIV, syphilis, gonorrhea, and chlamydia. Subsequently, participants will complete a quantitative survey—including a social and sexual network inventory—and an exit interview to review test results and confirm participants' contact information. Individual, interpersonal, and environmental factors will be assessed at baseline and follow-up visits (6 and 12 months). The primary outcomes are HCV prevalence and incidence. Secondary outcomes are sexual behavior, substance use, and psychosocial health.

currently ongoing. Once data have been collected, all data will be made publicly available.

**Funding:** This project was funded by Gilead Sciences, Inc (Grant Number: IN-US-987-5568). The funders had no role in study design, data collection and analysis, decision to publish, or preparation of the manuscript.

**Competing interests:** The authors have declared that no competing interests exist.

**Abbreviations:** HIV, Human immunodeficiency virus; HCV, Hepatitis C; CDC, Centers for Disease Control and Prevention; SMM, Sexual minority men; PWH, People with HIV.

## Results

To date (March 2023) a total of 162 participants have completed baseline visits at the DC study site and 161 participants have completed baseline visits at the Texas study site.

## Conclusion

This study has several implications that will directly affect the health and wellness of Black and Latino SMM. Specifically, our results will inform more-focused HCV clinical guidelines (i.e., effective strategies for HCV screening among Black/Latino SMM), intervention development and other prevention and treatment activities and development of patient assistance programs for the treatment of HCV among uninsured persons, especially in Deep South, that have yet to expand Medicaid.

## Introduction

New cases of acute Hepatitis C (HCV) have increased rapidly in the United States (U.S.) since 2010 [1]. In 2019, the Centers for Disease Control and Prevention (CDC) estimated 57,500 new cases of HCV infections in the U.S., representing a 14% increase from the previous year [2]. Stark racial/ethnic disparities exist in HCV infection in the U.S., with higher rates observed among Black and Latino populations [3–5]. An analysis of National Health and Nutrition Examination Survey data collected between 2001 and 2010 found that the prevalence of HCV antibodies was higher among non-Hispanic Black populations (2.2%) than non-Hispanic Whites (1.3%), demonstrating higher rates of prior HCV infection among non-Hispanic Black populations [6]. These disparities are further reflected in national HCV mortality rates where between 1995 and 2004, there was a 170% mortality rate increase among non-Hispanic Black populations compared to 124% among non-Hispanic Whites [7]. While there is robust documentation of the disparity in HCV prevalence and mortality rates by race/ethnicity, there remains a dearth of research on the burden of HCV among populations with intersecting marginalized identities who might be most vulnerable for HCV, particularly Black and Latino sexual minority communities.

HCV is primarily spread through contact with infected blood, most often through the sharing of needles during intravenous drug use [8]. Recently, various studies have demonstrated that sexual activity may be a major driving force for HCV acquisition and transmission among sexual minority men (SMM) [9–13]. Sexual activity during drug use (crystal methamphetamine), fisting, condomless receptive anal sex with ejaculation, inconsistent condom use, engaging in group sex, recent syphilis or gonorrhea infection, and use of nasal-administered drugs (powdered cocaine) have all been associated with HCV transmission and acquisition among SMM [14–16]. However, the generalizability of these study findings are limited due to the cross-sectional study design, convenience sampling approach, and the participants being predominately or all white SMM, in large metropolitan cities in the U.S. (New York and San Francisco).

People with HIV (PWH) have an increased risk of HCV coinfection [17–19]. A recent meta-analysis that investigated the prevalence and incidence of HCV among SMM globally found a pooled HCV prevalence of 2% among SMM who were HIV-negative and 5% among SMM living with HIV [20]. A racially diverse, cross-sectional study of 1,028 SMM in New York City found that 8% of SMM living with HIV were co-infected with HCV, compared with

2% of HIV-negative SMM [21]. The high prevalence of HIV among Black and Latino SMM [22] suggests that this group may be disproportionately vulnerable to HCV infection, compared to white SMM.

HIV and HCV coinfection have been well investigated and documented [23]. However, data on the prevalence and incidence of HCV among HIV-negative SMM are scarce. To date, only a few studies have examined HCV rates and corresponding risk factors among HIV-negative SMM, and there is a prominent gap in the literature for racial and ethnic SMM groups. The *Together 5000* study (a national longitudinal study of HIV-uninfected cisgender men, transgender women, and transgender men who have sex with men in the U.S.) found that 27% of the sample reported risk factors for HCV infection, such as: history of incarceration, prior use of HIV pre- or post-exposure prophylaxis, history of HIV testing, and recent methamphetamine use [24]. While these studies have filled a gap in the literature on the prevalence of HCV and correlates among SMM, these studies were cross-sectional and conducted almost exclusively with White SMM in the northeastern U.S.

To our knowledge, no previous empirical research study has prospectively investigated HCV among HIV-negative Black and Latino SMM. Consequently, this first-of-its-kind study aims to quantify the prevalence and incidence of HCV and determine the multilevel factors associated with HCV infection among HIV-negative, Black and Latino SMM in the Southern U.S. The two specific aims that guide this study are: (1) determine longitudinal relationships between sexual and drug use behaviors and HCV infection among a sample of 400 HIV-negative, Black and Latino SMM in the greater Washington, DC and Dallas, TX area and (2) characterize associations between social/sexual network characteristics and HCV infection. Our hypotheses are (1) sexual and drug use behaviors will be significantly associated with HCV prevalence and incidence, and (2) increased sexual network size, inconsistent condom use, and unknown partner HIV status, will be associated with higher HCV prevalence and incidence.

## Methods

### Study sites and target population

The present study builds upon a long-standing collaborative partnership between two community-based, non-profit organizations: Us Helping Us, People Into Living, Inc., (Washington, DC: www.ushelpingus.org) and Abounding Prosperity, Inc., (Dallas, TX: www.aboundingprosperity.org). These premiere Black SMM-led organizations have over 50 years of experience providing medical and social services to improve the health and wellbeing of Black and Latino SMM.

Us Helping Us, People into Living, Incorporated (Us Helping Us) is a non-profit organization founded in 1985, with the mission to improve the health and well-being of Black SMM through innovative programs and services and–through a vision of inclusiveness–to reduce the impact of HIV/AIDS in the entire Black community. This organization is renowned for its work at the intersection of healthcare service delivery, community-based education and mobilization, advocacy and research. Us Helping Us currently implements and manages HIV/STI prevention, treatment, and care programs, including via telehealth modalities (e.g., at-home HIV self-testing), behavioral health programs and services, community education, and community-led support and social groups designed to address the complex and multi-dimensional needs of highly marginalized communities, especially Black SMM, transgender and cisgender women, and people who use drugs. Us Helping Us accomplishes its mission through an integrated case management model, patient navigation, behavioral/mental health, primary care, social services; high-impact public health and prevention programming, community-based

research and advocacy, psychosocial support groups, and community education and mobilization efforts.

Abounding Prosperity, Inc. (AP, Inc.) was founded in 2005, with the mission to provide services that address health, social and economic disparities among Black Americans with a particular emphasis on gay & bisexual men, cisgender women, transgender women, and their families. The programs and services offered by the organization include strategies for mobilizing and organizing public institutions to effectively work together, with members of the impacted community to increase knowledge through a comprehensive plan that includes health education and disease prevention.

The collaboration between both organizations was fostered by previous programmatic collaborations in the past that laid the groundwork for research collaboration. Notably, both organizations have community-academic partnerships with local institutions, including Howard University and George Washington University (DC) and Texas Southwestern (Dallas), which propagates collaboration opportunities and sharing of resources. Additional programmatic support is added by the availability of mobile health units which both organizations will utilize to conduct baseline visits for the current study.

## Availability of study participants

The population source for this study is Black and Latino SMM in the Washington, DC and Dallas, TX metropolitan areas respectively. To adequately power the study and account for the lower prevalence of HCV infection among HIV-negative SMM suggested by previous literature, this study will enroll 400 Black and Latino SMM (200/site). The sample size was calculated using simple random sampling and our sample size of 400. The study has been designed so that each study site will enroll approximately 100 Black and 100 Latino SMM.

## Sample size

Estimating 85% retention, we propose to enroll a sample of 400 HIV-negative, Black and Latino, cisgender male participants in Washington DC and Texas. The primary outcomes are HCV prevalence and incidence. A sample of 400 participants provides statistical power (with 95% confidence and 80% power) to detect scientifically significant relative differences of 10%, 15%, and 20% in each of these outcomes between the two study sites. Using the methods described by Rosner for sample size estimation for longitudinal studies [25], a sample size of 200 participants in each group will ensure statistical power (using a two-sided two-sample t-test) to identify differences in HCS prevalence and incidence.

Participants at the Washington, DC site will be recruited from urban and suburban communities in the Washington-Arlington-Alexandria metropolitan area. This area, which includes the District of Columbia and portions of Virginia and Maryland, has a population of over 6.2 million people. A quarter of the area's population identify as Black or African American and 16% identify as Hispanic or Latino. The per capita income in Washington, DC is $51,437.

Similarly, participants at the Dallas, TX site will be recruited from the eleven-counties of Collin, Dallas, Denton, Ellis, Hunt, Kaufman, Rockwall, Johnson, Parker, Tarrant, and Wise. Of the estimated 7.5 million people who live in this area, 16% identify as Black or African American and 29% identify as Hispanic or Latino. The per capita income in Dallas, TX is $36,274.

Black and Latino SMM are estimated to make most of the SMM in Washington, DC and Texas. In Washington, DC, Black SMM make up almost half (43%) of all SMM while Latino SMM account for 10% [26]. In Texas, Latino SMM make up a third of all SMM while Black

SMM make up 9% [26]. The Dallas, TX and Washington, DC metropolitan area constitute the first and third most populous metropolitan statistical areas (MSAs) in the southern U.S., providing ample populations to identify and recruit Black and Latino SMM. Additionally, research has demonstrated Washington, DC and Texas to have high estimated HCV prevalence compared to the national average. A recent study found that Washington, DC had the highest estimated HCV prevalence in the U.S. while Texas had the 12th highest prevalence [5]. The prevalence of HCV among SMM Washington, DC and Dallas, Texas are unknown and this program aims to fill this knowledge gap.

## Recruitment

Recruitment began in September 2021 and continue until recruitment targets are met. Recruitment will begin at the Dallas, TX site after any issues in recruitment and study protocol implementation identified at the Washington, DC site are resolved. Participants will be recruited through: 1) advertisement through flyers and posters in the offices of partner community-based organizations, 2) outreach at gay nightclubs and bars and community events such as DC and Dallas Southern Pride, 3) geospatial social/sexual networking smartphone applications (e.g. Grindr, Jack'd), 4) our ongoing high-impact HIV prevention interventions, 5) word of mouth referrals, and 6) snowball sampling.

## Inclusion/Exclusion criteria

Participation in this study is open to individuals ages 18 years and over and identify as Black/African American or Latino. Inclusion criteria further require that individuals: (1) self-identify as biological male at birth and currently identify themselves as male, (2) live in the Washington, DC or Dallas, TX metropolitan areas, and (3) have had oral or anal sex with a man in the six months before enrollment. Additionally, participants will be confirmed HIV-negative at their baseline visit. Exclusion criteria were: 1) being younger than 18 years of age, we limited participants to this age due to some of the sensitivity of the survey especially related to psycho-social health; 2) participants self-identifying their race as anything other than African American or Black or their ethnicity other than Latino or Hispanic; 3) participants reporting their current residence outside of Washington, DC or Dallas, TX MSA; 4) Self-identifying as a cis-gender or transgender woman; 5) reporting having sex with women exclusively; 6) confirmed to be living with HIV/AIDS; 7) being unable to read or speak English or Spanish;8) being unable to complete the follow-up study visits.

## Ethics approval

The Sterling Institutional Review Board reviewed and approved this study in accordance with federal regulations for research with human subjects.

## Screening procedures

Research staff will screen interested individuals for enrollment in the study with a brief questionnaire. To determine eligibility, self-reported telephone or in-person screenings will be completed. These screenings will be completed quickly (3–5 minutes) and individuals deemed eligible will be scheduled for an enrollment assessment visit at the respective study site. Each appointment will be scheduled in two-hour blocks to allow time for completion of all study procedures and administrative tasks (e.g., informed consent process). Research staff will call or text each participant the day before their scheduled appointment and at least one hour before to confirm or reschedule the appointment.

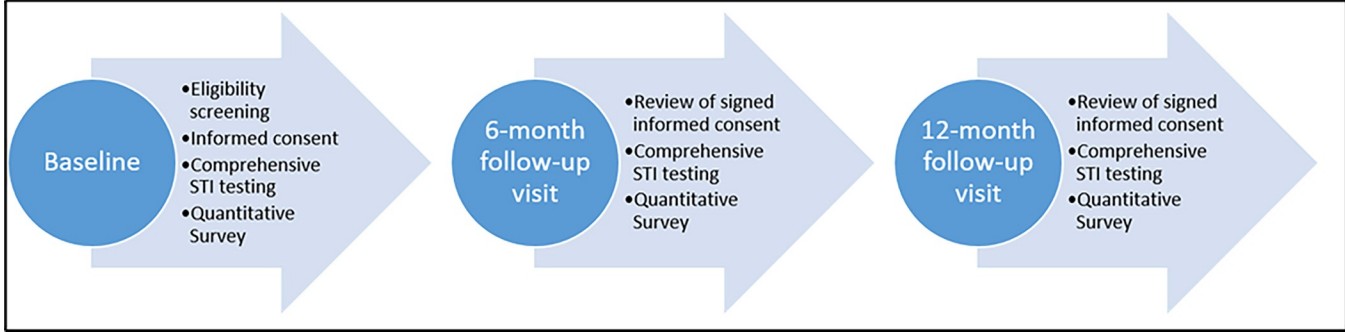

**Fig 1. Illustration of the study flow from baseline to follow-up visits.**

## Overview of the study visits

The initial study visit will begin with the informed consent process, followed by a series of health screenings and a quantitative questionnaire (Fig 1). We estimate that each visit will take approximately 1–1.5 hours to complete. Participants will meet with study personnel in a private room at each study location to review the informed consent form. They will be provided a study synopsis, possible harms and benefits associated with participation, and IRB contact information. Individuals who agree to participate in the study will provide written informed consent and will be given a copy of the consent document for their records. Following the informed consent process, participants will undergo a series of screenings, including rapid testing for HIV, HCV, syphilis, and provide self-collected urine, pharyngeal and rectal samples for chlamydia and gonorrhea testing. Finally, participants complete an electronic study questionnaire. These procedures are described in greater detail below.

## Integrated sti screening

Participants will undergo rapid HIV, HCV, and syphilis tests using blood samples obtained via finger prick. Next, participants will be escorted to a private restroom and instructed to perform a self-collected urine specimen and pharyngeal and rectal swab for gonorrhea and chlamydia testing. Previous studies have supported the reliability of self-collected swabs for detecting gonorrhea and chlamydia among SMM [27]. Participants will be made aware of their rapid test results during their visit. A community health worker will follow-up with each participant regarding their gonorrhea and chlamydia test within 72 hours. Participants with a reactive rapid HIV, HCV, or syphilis test will be immediately linked to an accessible contract laboratory for confirmatory testing. Participants with confirmed infections will be linked to the appropriate treatment. A confirmed HIV infection will result in exclusion from study participation.

## Study questionnaire

After receiving a negative HIV rapid test and completing the HCV and STI testing, participants are escorted to a private interview room to complete a computer-assisted questionnaire containing scales to measure the proposed correlates of HCV infection outlined in our aims/hypotheses. A complete list of measures is provided in Table 1. We selected measures that characterize the multilevel and complex amalgam of biological and social, behavioral and environmental factors that will typify the HCV 'riskscape', as described by previous literature. These include sexual heath and behavior question including knowledge of HCV status, perceived risk and awareness of HCV. We also assessed for various psychosocial health outcomes

**Table 1. Domains of question assessed at various study timepoints.**

| | Measure | Time of assessment | | |
|---|---|---|---|---|
| | | Baseline | 6 months | 12 months |
| **Sociodemographic characteristics** | Age, income, race, sexual orientation, socioeconomic status, housing, health insurance immigration status | X | X | X |
| **Sexual Health/Behaviors** | Hepatitis C status and awareness | X | X | X |
| | Sexual History and experiences, including condom use and fisting practices | X | X | X |
| | HIV Testing / PrEP Knowledge and Use | X | X | X |
| | Perceived Risk of Hepatitis C | X | X | X |
| | Perceived Risk of HIV [28] | X | X | X |
| | Alcohol and Drug use (before and during sexual activity) | | X | X |
| | Intimate partner violence (physical, emotional, and sexual) [29] | X | X | X |
| | Healthcare utilization | X | X | X |
| **Psychosocial Health** | Depressive symptoms [30] | X | X | X |
| | Social Isolation [31] | X | X | X |
| | Suicidality [32] | | X | X |
| | Adverse Childhood Experiences [33] | | X | X |
| | Resilience [34] | | X | X |
| | Everyday Discrimination Scale [35] | | X | X |
| | Social Networks and Support (Friends, Relatives, Neighbors) | | X | X |
| | Happiness [36] | | X | X |
| | Life Satisfaction [37] | | X | X |
| | Courage [38] | | X | X |
| | Quality of Life [39] | | X | X |
| | Self Esteem [40] | | X | X |
| **Networks** | Social And Sexual Network Inventory | X | X | X |
| | Communication with sexual partners | X | X | X |

such as depressive symptoms, suicidality, adverse childhood experiences, and others (Table 1). Additionally, we included novel measures that have not been included in previous research to capture data relating to potential confounders and effect modifiers of our hypothesized associations. All questions are programmed on a cloud-based survey tool and administered via a computer tablet.

Finally, participants will meet with study staff at the end of each visit to ensure that all tests and surveys are complete, confirm the participants' contact information, and schedule their follow-up visits (6 months and 12 months). Participants will be compensated $50 cash for their time and given a bag of quality condoms and lubricants at each study visit.

## Translation of study materials

All study materials will be made available both in English and Spanish. We undertook an iterative, culturally-informed process to ensure that Spanish-speaking participants would easily understand the study materials.

## Analysis plan

We will compute descriptive statistics (means, standard deviations, and proportions) for all study variables. Next, we will run logistic regression and cox proportional hazards models to examine the association between multilevel factors and HCV prevalence (testing conducted at

baseline) and incidence (testing conducted during 6 and 12-month follow-up visits), adjusting for potential confounders.

Additionally, we will fit generalized estimating equation (GEE) models to examine the association between sexual network characteristics and HCV infection outcomes over time. We will also perform dynamic network analyses using two main types of models. The first involves summarizing network change from time to time via a specific measure and then modeling using standard models within the generalized linear model (GLM) framework [41]. The resulting logistic regression will allow us to estimate how respondent-level characteristics are related to the likelihood of relationship dissolution; fitting a conditional logistic model to the same outcome will allow exploration of the association between alter and/or relationship-level characteristics and the likelihood of dissolution [42]. We will conduct the above analysis stratified by site to assess for any study site specific differences.

## Data management, quality assurance and control, and statistical analysis

Us Helping Us is the coordinating center for the research study and is responsible for all data management, quality assurance and control procedures, and data analyses activities. In collaboration with the Dallas site research team, Us Helping Us will maintain a comprehensive data management system including quality assurance and control procedures to ensure rigorous and high-quality data collection and adherence to study protocols. All study-related documents, including signed copies of the informed consent form, will be stored in locked cabinets at each community-based organization. The study researchers (including the PI and co-Investigators) will each have completed the NIH's Protection of Human Subjects training course (or an acceptable equivalent). Safeguards in these spaces include: (a) password-protected computers; (b) screen savers with password required re-entry on computers automatically initiate after 20 minutes of nonuse; (c) all computers are located in positions and angled so that casual observers cannot read computer screens; (d) all computers contain the most current virus and security software and receive regular updates of this software; (e) all data collection materials are stored in locked filing cabinets in locked offices and access is limited to the PI. In all cases, data from the study will only be accessible by the PI and a limited number of study investigators.

## Dissemination plan and data sharing

We expect that novel findings from this study and will have immense public health significance and applicability. The research team has devised a detailed plan to disseminate study findings to key stakeholders and to circulate study data to study collaborators. The dissemination plan includes conducting presentations at academic and community meetings, publication of study findings in top peer-reviewed journals, and production of summary sheets of published work to be circulated to study participants and community agencies that serve SMM. Additionally, we will develop a research writing group, that will convene monthly to discuss preliminary study results, develop concept sheets for additional scientific manuscripts, and workshop follow-up studies that will be informed by study findings. Lastly, external researchers will have the opportunity to submit concept sheets for manuscript development. Study investigators also maintain a Data Distribution Agreement in order to protect the privacy and confidentiality of participants. Researchers interested in accessing study data may contact the principal investigator and must agree to adhere to the requirements of the data distribution agreement.

## Discussion

The current study will investigate the longitudinal association between sexual and drug use behaviors, sexual network characteristics and HCV infection among a sample of 400 HIV-

negative, Black and Latino SMM in Washington, DC and Dallas, T.X. The results of this research study will have several implications that directly affect the health and wellness of Black and Latino SMM. First, these findings will inform clinical procedures around the need for routine testing for HCV and screening for sexual behavior that might increase the likelihood of HCV infection. Second, we plan to develop evidence-based and culturally-relevant interventions to ultimately increase knowledge and HCV testing behavior and decrease rates of HCV infection among Black and Latino SMM. Lastly, we hope to conceptualize further research studies related to HCV and other social/structural determinants of health among Black and Latinx SMM and their mechanisms of action.

## Study limitations

There are several study limitations worth highlighting. First, the current study is specific to Black and Latino SMM in two regions of the U.S., therefore the study findings may not be generalizable to other SMM. Second, some of our results will rely on self-reported data, which is vulnerable to social desirability and recall bias. Third, our recruitment approach might exclude Black and Latino SMM who don't receive health services at our partner organizations, frequent LGBT social spaces, or within the social networks of our enrolled participants. The exclusion of Black and Latino not reached through our recruitment efforts may potentially bias our findings especially related to access to healthcare services and providers.

## Study strengths

Despite the limitations outlined, this study has several strengths. First, the longitudinal study design allows us to draw conclusions related to causation rather than just association. Second, the multi-site and multiracial study design allows to conduct comparison in study effects across racial and geographical categorizations. Third, the iterative approach to study materials translation from English to Spanish will ensure that the data been collected is sound and culturally grounded. Lastly, we believe the specific focus on Black and Latino SMM is a major strength of this study as they are disproportionately impacted by infectious diseases including HCV and have unique needs and lived experience that impact their ability to assess testing and treatment.

## Acknowledgments

We appreciate all individuals who have and will participate in the study.

## Author Contributions

**Conceptualization:** Temitope Oke, Omar Martinez, DeMarc A. Hickson.

**Data curation:** DeMarc A. Hickson.

**Formal analysis:** Adedotun Ogunbajo, DeMarc A. Hickson.

**Funding acquisition:** Omar Martinez, DeMarc A. Hickson.

**Investigation:** Omar Martinez, DeMarc A. Hickson.

**Methodology:** Omar Martinez, Carl Latkin, DeMarc A. Hickson.

**Project administration:** Adedotun Ogunbajo, Mitchell Brooks, Omar Martinez, Kirk Myers, DeMarc A. Hickson.

**Supervision:** DeMarc A. Hickson.

**Writing – original draft:** Adedotun Ogunbajo.

**Writing – review & editing:** Adedotun Ogunbajo, Mitchell Brooks, Temitope Oke, Omar Martinez, Carl Latkin, Kirk Myers, DeMarc A. Hickson.

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
