## [Decision Letter · Decision Letter 0]

13 May 2022

PONE-D-22-09423Hepatitis C (HCV) among Black and Latino sexual minority men (SMM) in the Southern United States: Protocol of an epidemiological studyPLOS ONE

Dear Dr. Ogunbajo,

Thank you for submitting your manuscript to PLOS ONE. After careful consideration, we feel that it has merit but does not fully meet PLOS ONE’s publication criteria as it currently stands. Therefore, we invite you to submit a revised version of the manuscript that addresses the points raised during the review process. The reviewers had excellent suggestions. The sample size calculation should use a method realistic for the method of analysis. One source to consider for sample size estimation. There are a number of vignettes on the internet demonstrating how to do these calculations in practice.

Gelman A, Carlin J. Beyond Power Calculations: Assessing Type S (Sign) and Type M (Magnitude) Errors. Perspect Psychol Sci. 2014 Nov;9(6):641-51. doi: 10.1177/1745691614551642. PMID: 26186114.

We look forward to receiving your revised manuscript.

Kind regards,

Janet E Rosenbaum, Ph.D.

Academic Editor

PLOS ONE

Journal Requirements:

Reviewers' comments:

Reviewer's Responses to Questions

**Comments to the Author**

1. Does the manuscript provide a valid rationale for the proposed study, with clearly identified and justified research questions?

Reviewer #1: Yes

Reviewer #2: Yes

2. Is the protocol technically sound and planned in a manner that will lead to a meaningful outcome and allow testing the stated hypotheses?

Reviewer #1: Yes

Reviewer #2: Yes

3. Is the methodology feasible and described in sufficient detail to allow the work to be replicable?

Reviewer #1: Yes

Reviewer #2: Yes

4. Have the authors described where all data underlying the findings will be made available when the study is complete?

Reviewer #1: No

Reviewer #2: No

5. Is the manuscript presented in an intelligible fashion and written in standard English?

Reviewer #1: Yes

Reviewer #2: Yes

6. Review Comments to the Author

You may also provide optional suggestions and comments to authors that they might find helpful in planning their study.

Reviewer #1: Overall the study protocol is clear and concise, rationale and hypotheses of the study are well structured. However, before acceptance, I suggest that the auhors make changes to the points below for clarity to some significant aspects:

Major revisions:

1. The inclusion of 400 participants, how was this defined as enough? did the authors do sample size calculation? If yes (or not) this should be made clearer in the protocol.

2. HCV confirmation status during follow-up (month 6 and 12), how will this be done? I'm not clear whether it will only be self-reported by the participants through questionniare or that an additional laboratory test will also be done?

3. Study questionnaire: have the baseline and follow-up questionnaires been developed? If yes they can be included as supplementary materials for better understanding.

4. Analysis plan: How incidence and prevalence of HCV will be distinguished? it is not clear. This is important to explain to reader about the calculation of incidence rates or prevalence rates.

Also, how will authors determine the date of incidence? through laboratory test date? mid-point methods? the gap between follow-up is relatively long (six months) in this study, therefore, the right method to define when incident happen is important.

Minor revisions / suggestions:

1. Typos in wording 'latinx' instead of latino found in some paragraphs, please correct them.

2. Title and methods: the authors use 'epidemiological' study. I would change it to 'prospective' study instead (epidemiological studies also include cross-sectional, ecological, etc).

3. It would be good to have a flow chart or diagram figure that shows the study phases from baseline until follow-up, including the procedures in each visit.

Thank you

Reviewer #2: This protocol paper describes a study designed to evaluate factors associated with Hepatitis C infection among HIV-negative Black and Latino sexual minority men. The topic is of critical importance and is currently underrepresented in the literature. Overall, the paper is well written and quite clear. There are a few areas that could benefit from additional detail or minor revision.

1.) Study design: What is the target sample size? Has a power analysis been completed?

2.) Study measures: Please add a time component to each measure. For example, will drug and alcohol use be assessed over last year, last month?

3.) Analysis plan: Logistic regression is not the best method to evaluate relationships with common outcomes. Odds ratios only approximate risk when the outcome is sufficiently rare. Thus, strongly recommend using Modified Poisson or log binomial regression if the outcome affects greater than 10% of the study population.

4.) Analysis plan: How will the snowball sampling be accounted for in analyses?

7. PLOS authors have the option to publish the peer review history of their article (what does this mean?). If published, this will include your full peer review and any attached files.

Reviewer #1: No

Reviewer #2: No

---

## [Decision Letter · Decision Letter 1]

24 Jan 2023

PONE-D-22-09423R1Hepatitis C (HCV) among Black and Latino sexual minority men (SMM) in the Southern United States: Protocol of a prospective cohort epidemiological studyPLOS ONE

Dear Dr. Ogunbajo,

Thank you for submitting your manuscript to PLOS ONE. After careful consideration, we feel that it has merit but does not fully meet PLOS ONE’s publication criteria as it currently stands. Therefore, we invite you to submit a revised version of the manuscript that addresses the points raised during the review process. Please address all comments by the reviewers.

We look forward to receiving your revised manuscript.

Kind regards,

Janet E Rosenbaum, Ph.D.

Academic Editor

PLOS ONE

Journal Requirements:

Reviewers' comments:

Reviewer's Responses to Questions

**Comments to the Author**

1. Does the manuscript provide a valid rationale for the proposed study, with clearly identified and justified research questions?

Reviewer #3: Yes

Reviewer #4: Yes

Reviewer #5: Yes

2. Is the protocol technically sound and planned in a manner that will lead to a meaningful outcome and allow testing the stated hypotheses?

Reviewer #3: Yes

Reviewer #4: Partly

Reviewer #5: Yes

3. Is the methodology feasible and described in sufficient detail to allow the work to be replicable?

Reviewer #3: Yes

Reviewer #4: No

Reviewer #5: Yes

4. Have the authors described where all data underlying the findings will be made available when the study is complete?

Reviewer #3: Yes

Reviewer #4: Yes

Reviewer #5: Yes

5. Is the manuscript presented in an intelligible fashion and written in standard English?

Reviewer #3: Yes

Reviewer #4: Yes

Reviewer #5: Yes

6. Review Comments to the Author

You may also provide optional suggestions and comments to authors that they might find helpful in planning their study.

Reviewer #3: The authors describe a study protocol for Hepatitis C among Black and Latino sexual minority men in the South. Overall the planned study is interesting, informative, and addresses an important research question. It appears to be improved based on the tracked revision as well. I only have a few minor recommended revisions for clarity.

1. In general, I recommend not using the term "Blacks", instead using "Black populations", "Black participants", "Black individuals", etc.

2. Have any participants been recruited at the Texas site? Only the number for the DC site is currently specified.

3. I recommend a brief sentence in the discussion on how the recruitment limitations (i.e., excluding Black and Latino SMM who don’t receive health services) may potentially bias results (such as leading to an overestimate of healthcare-access related factors).

4. I recommend adding the focus on an important minority population for HCV prevention/treatment efforts as a strength.

Reviewer #4: 1. The study fails to adress how 400 participants is large enough to answer the research question and to draw valid conclusions. The authors should rewrite this section and include the sample size calculation, taking into account an assumed drop-out rate if applicable.

2. While the written text appears to be sound, the captions of the figure and table are unclear or incorrectly used, making it difficult to understand the figure and table on its own. I would suggest using a more thorough and complete caption that really describes what the figure or table is about. For instance for the figure I would suggest using something like "Illustration of the study design". For the table, I would suggest using a caption like "Outcomes collected in the study". In addition, although most of the outcomes seem to be included in the table and the table is of added value to the manuscript, the table fails to give an immediately clear overview when certain outcomes are collected. I would advise doing something like this to make it immediately and easily clear to the reader:

Baseline Month 6 Month 12

Sociodemographics

Age X X X

Income X X X

Lastly, I believe some collected outcomes are missing in the table (e.g. STI testing). In the text of the manuscript, nothing is being said about the evaluation of psychosocial health outcomes, but you are collecting information about this. The authors should include more detail about why this information is collected.

3. Although the authors clearly describe the inclusion criteria, the exclusion criteria remain unclear and should be incorporated in the manuscript.

4. I am not a statistician, so I can not completely review the statistical analyses you have planned. However, I believe some valuable information is missing about interim or subgroup analysis. Are the authors planning to do so or why not. And if yes, why would the authors do that.

5. The study fails to address confidentiality of the data being collected. The authors should described who has access to the data, if data is managed anonymously or is coded.

6. The authors are off to a good start, however, a large number of references used are quite old (from 2013 or before). The authors should include more up-to-date references to make their research stronger.

Reviewer #5: The manuscript discusses a study protocol that seeks to inform HCV clinical guidelines (i.e., effective strategies for HCV screening among Black/Latino SMM), intervention development, prevention and treatment activities and development of patient assistance programs for the treatment of HCV among uninsured persons in the U.S. Deep South.

Throughout, the authors discuss the employment of a prospective cohort epidemiological study of HCV among Southern U.S. Black and Latino sexual minority men. The proposed study sample would include 400 Black and Latino SMM, aged 18 years and above. At the time of manuscript submission, 92 participants had been enrolled in study at the Washington D.C. site.

Introduction. The manuscript includes a strong introduction and literature review. The epidemiological data presented is clear, timely, and nuanced. In particular, the authors note the dearth of current scholarship focused on HCV burden among intersectionally marginalized communities, such as Black and Latino sexual minority men. Overall, both the introduction and literature review provide solid justification for the proposed study protocol and empirical study of

HCV among HIV-negative Black and Latino SMM.

Methods: Study sites and target population. While the methods section, including the subsection 'Study Sites and Target Population,' is well-crafted, in describing the ongoing collaborative partnership between the two key community-based, non-profit organizations, the authors should describe how their partnerships with the community organizations were facilitated. Presently, this is unclear.

Methods: Sample Size. In detailing the proposed protocol study’s research design, the authors should provide an empirical rationale for the proposed study sample size (N = 400). At present, no clear justification is provided.

Methods: Inclusion/Exclusion Criteria: Age.

The reviewer strongly suggests that the authors provide an empirical rationale for the stated inclusion/exclusion criteria related to age (i.e., 18 years and up). Indeed, 18-years and over encompasses a range of developmental stages that may be related to HCV incidence and/or prevalence among the population of interest (BLSMM). As mentioned, please provide an empirical rationale for this methodological relative to other studies that focus on particular age groups, e.g., ages 13 to 24; 18 to 25; 30 and up. Given that the manuscript is a study protocol, it is important for the authors to unpack their methodological decision-making in its entirety.

Methods: Overview of Study Visits. Notably, the reviewer appreciates the careful attention paid to unpacking this component of the proposed research protocol (i.e., Initial study visit, Integrated STI screening, Study questionnaire, Instruments, Translation of study materials, analysis plan, data management, quality assurance, & control, and statistical analysis). Important given the outcomes and study population.

7. PLOS authors have the option to publish the peer review history of their article (what does this mean?). If published, this will include your full peer review and any attached files.

Reviewer #3: No

Reviewer #4: No

Reviewer #5: No

---

## [Decision Letter · Decision Letter 2]

20 Jun 2023

Hepatitis C (HCV) among Black and Latino sexual minority men (SMM) in the Southern United States: Protocol of a prospective cohort epidemiological study

PONE-D-22-09423R2

Dear Dr. Ogunbajo,

We’re pleased to inform you that your manuscript has been judged scientifically suitable for publication and will be formally accepted for publication once it meets all outstanding technical requirements.

Kind regards,

Janet E Rosenbaum, Ph.D.

Academic Editor

PLOS ONE

Additional Editor Comments (optional):

Reviewers' comments:

Reviewer's Responses to Questions

**Comments to the Author**

1. Does the manuscript provide a valid rationale for the proposed study, with clearly identified and justified research questions?

Reviewer #4: Yes

Reviewer #5: Yes

2. Is the protocol technically sound and planned in a manner that will lead to a meaningful outcome and allow testing the stated hypotheses?

Reviewer #4: Yes

Reviewer #5: Yes

3. Is the methodology feasible and described in sufficient detail to allow the work to be replicable?

Reviewer #4: Yes

Reviewer #5: Yes

4. Have the authors described where all data underlying the findings will be made available when the study is complete?

Reviewer #4: Yes

Reviewer #5: Yes

5. Is the manuscript presented in an intelligible fashion and written in standard English?

Reviewer #4: Yes

Reviewer #5: Yes

6. Review Comments to the Author

You may also provide optional suggestions and comments to authors that they might find helpful in planning their study.

Reviewer #4: I thank the authors for the updated version of this manuscript, this is a way more extensive version and easier to understand. All the comments and suggestions that I previously submitted have been included.

Reviewer #5: The authors have sufficiently addressed this reviewer's comments. The manuscript is sufficient for publication in PLOS One.

7. PLOS authors have the option to publish the peer review history of their article (what does this mean?). If published, this will include your full peer review and any attached files.

Reviewer #4: No

Reviewer #5: No

---

## [Editor Report · Acceptance letter]

26 Jun 2023

PONE-D-22-09423R2 

Hepatitis C (HCV) among Black and Latino sexual minority men (SMM) in the Southern United States: Protocol of a prospective cohort epidemiological study 

Dear Dr. Ogunbajo:

I'm pleased to inform you that your manuscript has been deemed suitable for publication in PLOS ONE. Congratulations! Your manuscript is now with our production department. 

Kind regards, 

on behalf of

Dr. Janet E Rosenbaum 

Academic Editor

PLOS ONE